# Oncological and Functional Outcomes of Hemi-Ablation Versus Focal Ablation for Localized Prostate Cancer Using Irreversible Electroporation

**DOI:** 10.3390/cancers17132084

**Published:** 2025-06-22

**Authors:** Michel Suberville, Kai Zhang, Jean Baptiste Woillard, Isabelle Herafa, Dorothée Ducoux, Rachid Nachef, Jeremy Teoh, Gang Zhu, Chi-Fai Ng, Pilar Laguna, Jean de la Rosette

**Affiliations:** 1Pôle Saint German Centre Hospitalier, 19100 Brive la Gaillarde, France; michel.suberville@ch-brive.fr; 2Department of Urology, Beijing United Family Hospital and Clinics, Beijing 100015, China; zhu.gang@ufh.com.cn; 3Inserm U1248, University of Limoges, Department of Pharmacology and Toxicology, CHU de Limoges, 87000 Limoges, France; jean-baptiste.woillard@unilim.fr; 4Clinical Investigation Center, CHU de Limoges, 87000 Limoges, Francedorothee.ducoux@chu-limoges.fr (D.D.); 5Department of Clinical Study, Center Hospitalier de Brive, 19100 Brive la Gaillarde, France; rachid.nachef@ch-brive.fr; 6S.H. Ho Urology Centre, Department of Surgery, The Chinese University of Hong Kong, Hong Kong 999077, China; jeremyteoh@surgery.cuhk.edu.hk (J.T.); ngcf@surgery.cuhk.edu.hk (C.-F.N.); 7International School of Medicine, Istanbul Medipol University, Istanbul 34810, Türkiye; plaguna@medipol.edu.tr (P.L.); j.j.delarosette@gmail.com (J.d.l.R.); 8Department of Uorlogy, Bashkir State Medical University, Ufa 450008, Russia

**Keywords:** irreversible electroporation, side effect, patient-reported quality of life, prostate cancer, focal therapy

## Abstract

For localized prostate cancer, focal therapy uses different sources of energy to treat selected areas of the gland, thereby avoiding or limiting damage to surrounding structures and aiming to preserve urinary and erectile function. Irreversible electroporation (IRE) is a novel tumor ablation technique using a non-thermal energy source, which might have potential advantages over other focal therapy modalities and achieve superior functional and safety outcomes. The aim of our retrospective study was to assess the oncological control, functional outcomes, and quality-of-life results of hemi-IRE ablation versus focal IRE ablation therapy for localized prostate cancer patients in a short- to medium-term follow-up. We confirmed that both IRE protocols achieved good urinary and erectile function outcomes and favorable short-term oncological control and that hemi-IRE ablation showed a significantly lower persistence of (clinically significant) prostate cancer compared with focal IRE ablation.

## 1. Introduction

Irreversible electroporation (IRE) is a novel treatment modality using pulsed, high-voltage, low-energy direct electric current for tumor ablation. For localized prostate cancer, active surveillance or whole-gland treatment including radical prostatectomy and radiotherapy are considered standard treatment options [1,2]. But whole-gland treatments are associated with considerable morbidity [3,4,5,6]. IRE and other focal therapy modalities were developed with the aim to minimize adverse effects while maintaining a beneficial oncological outcome [7,8,9,10,11].

As a non-thermal energy platform, IRE has shown favorable results, demonstrating a safe and effective focal treatment option with low patient morbidity, favorable functional outcomes, and good short-term oncological control [12,13,14,15,16]. However, the extent of IRE ablation and an optimal treatment protocol is still a point of discussion.

This study presents the results of hemi-IRE ablation versus focal ablation therapy for localized prostate cancer patients regarding oncological control, functional outcomes, and quality of life in a short- to medium-term follow-up.

## 2. Patients and Methods

### 2.1. Study Design

This is a retrospective, single-center study. The study was conducted according to good clinical practice and was IRB approved (23-2023-04). The patients received focal or hemi-ablation for their prostate cancer localized in one prostate lobe.

### 2.2. Patient Selection

The study population included patients with histologically confirmed prostate cancer. From January 2017 to June 2018, consecutive patients were treated with IRE according to a focal therapy protocol, and from June 2018 onwards, patients were consecutively treated with hemi-ablation. Informed consent was taken before the treatment.

### 2.3. Treatment Protocol

The NanoKnife System (AngioDynamics, Inc., Latham, NY, USA) was used. MRI-targeted fusion biopsy (MIM Symphony software, version 6.8.2, BK 5000 ultrasound scanner, MA, USA) plus systematic biopsy was performed to localize and diagnose the prostate cancer. Before biopsy, the patients received an mpMRI, and the mpMRI prostate measurements were entered into the NanoKnife planning software prior to the IRE treatment.

All the patients received single-shot iv Ofloxacin antibiotic prophylaxis. The electrodes were placed into the predefined ablation zone using biplane transrectal ultrasound image guidance to visualize both sagittal and axial views. In total, 90 consecutive high-voltage pulses (1500 V/cm) with a direct current between 20 and 50 A were delivered.

During the IRE procedure, the patients received muscle relaxants to prevent (severe) muscle contraction. The procedure was performed under general anesthesia, and all the patients received a suprapubic catheter. The patients were scheduled to stay overnight and were discharged the next day.

### 2.4. Study Outcomes

The primary outcome was persistence of prostate cancer at 1 year. The patients were scheduled for repeat MRI-targeted fusion biopsy plus systematic biopsy at 1 year post-IRE ablation. The secondary outcomes included functional outcomes measured by the International Prostate Symptom Score (IPSS) and International Index of Erectile Function-5 (IIEF-5) questionnaires.

### 2.5. Data Collection and Follow-Up

The procedure records, MRI, and biopsy results, including International Society of Urological Pathology (ISUP) grade groups and questionnaires, were collected. The patients were followed up at 3 months and every 6 months afterwards. Patients with a persistent tumor would receive subsequent treatment according to their biopsy results and based on shared decision-making.

### 2.6. Statistical Analysis

Descriptive statistics were used to present the different parameters during follow-up. The Mann–Whitney U test was performed to compare the IPSS and IIEF scores between the two groups. The chi-square test was used to compare the oncological results. A two-sided *p*-value of <0.05 was considered statistically significant. Statistical analyses were performed using SPSS for Windows (Version 27, IBM Corp., Armonk, NY, USA).

## 3. Results

### 3.1. Baseline Characteristics

From January 2017 to April 2023, 106 patients were recruited in this study, including 40 patients in the focal ablation group and 66 patients in the hemi-ablation group. The median follow-up time was 24 months (IQR 12–48). There were no statistically significant differences between the two groups regarding age, prostate-specific antigen (PSA), prostate volume, biopsy cores, and grade groups according to ISUP. Patients in the hemi-ablation group required significantly more IRE electrodes and longer suprapubic catheterization than patients in the focal ablation group. The demographics and baseline characteristics are presented in Table 1 and Figure 1.

### 3.2. Early Oncological Control

A total of 94 patients underwent repeat prostate biopsy at 12 months after IRE: 36 patients in the focal ablation group and 58 patients in the hemi-ablation group. Prostate cancer was found in 72.2% of the focal ablation group and in 31.0% in the hemi-ablation group (*p <* 0.001), and clinically significant prostate cancer (Gleason score ≥ 3 + 4) was detected in 25% of the focal ablation group and in 8.6% of the hemi-ablation group (*p* = 0.003).

In the focal ablation group, 44.4% of patients had a persistent tumor infield; 11.1% of patients had a persistent tumor outfield; and 16.7% of patients had persistent tumors both in- and outfield. In the hemi-ablation group, 8.6% of tumors were detected as infield, 20.7% as outfield, and 1.7% as in- and outfield.

Furthermore, 58.3% and 15.5% of patients underwent subsequent retreatment in the focal and hemi-ablation group, respectively. The treatment included radiotherapy, radical prostatectomy, second IRE, or androgen deprivation therapy. The results are summarized in Table 2 and Appendix A.

In total, 12 patients did not undergo repeat biopsy, including 4 patients in the focal ablation group and 8 patients in the hemi-ablation group. There was no significant difference between the patients with and without repeat biopsy regarding age, PSA, prostate volume, and ISUP distribution (Appendix A).

During follow-up, there was no prostate cancer-specific death or metastatic case.

### 3.3. Functional Outcomes

There were no significant differences in IPSS and IIEF between the focal ablation and hemi-ablation groups across all points in time. The IPSS score decreased in both groups. However, the IIEF in the focal ablation group increased during follow-up. In contrast, the IIEF declined in the hemi-ablation group. It decreased in the first 6 months and afterwards rose slightly but could not achieve the baseline level. The results on quality-of-life measures are summarized in Figure 1 and Appendix A.

## 4. Discussion

Hemi-ablation IRE therapy achieved significantly superior oncological results compared with focal ablation. Persistent significant prostate cancer was detected in 8.6% for the hemi-ablation group and in 25% for the focal ablation group. There were no statistically significant differences in the functional outcomes observed in the short–medium term.

IRE is a promising technique, aiming to only treat the tumor area to achieve oncological control whilst reducing treatment-related functional detriment. Several studies have reported the early favorable oncological results of IRE [16,17,18,19,20]. However, concern regarding IRE-focal therapy has centered on the knowledge that prostate cancer is multifocal in origin; residual tumor(s) can be detected both inside and outside the treated field following focal treatment. A systematic review displayed that, in the repeat biopsy, the infield clinically significant prostate cancer detection rate was between 0 and 33.3%, and the out-of-field clinically significant prostate cancer detected was between 0 and 30.8% [12]. In most studies, a repeat biopsy was conducted 6–12 months after IRE treatment. There is reason to believe that most tumors detected in the untreated area during repeat biopsy actually were the cancers missed in the first biopsy prior to IRE ablation.

Concerning the extent of the ablation area, it may be questioned whether it should be limited to the lesion, the region, or the lobe. In theory, an extended or a hemi-ablation could destroy a similar number of tumor lesions, including those that have not been identified and diagnosed in the mpMRI and biopsy before IRE. Our study bore out this hypothesis. Hemi-ablation resulted in a lower outfield clinically significant prostate cancer recurrence than that in focal ablation. In a multi-center, randomized study, the effect of focal versus extended IRE ablation on oncological control for low–intermediate risk prostate cancer was evaluated. In the repeat biopsy, prostate cancer was found in 56.3% of the focal ablation group and in 43.4% of the extended ablation group, and clinically significant prostate cancer (ISUP ≥ 2) was detected in 18.7% of the focal ablation group and in 13.2% of the extended ablation group. The extended IRE showed an improved recurrence rate over focal ablation, although the differences were not statistically significant [14]. It is interesting to note that, in our study, the infield persistence rate was much higher in the focal ablation group than that in the hemi-ablation group (61.1% vs. 10.3%). In a published study, the infield persistence rate was 16.7% vs. 20.8% for the focal and extended groups [14]. A potential explanation could be the too-small ablation area in the focal IRE group, which might not overcome the tumor margin. That was why we abandoned focal ablation and subsequently switched to hemi-ablation therapy.

Regarding the improved tumor persistence rate in the hemi-ablation group, it may be doubted whether this is due to the bias of a learning curve; after all, the hemi-ablation group patients underwent treatment after the focal ablation group patients. However, the learning curve for IRE treatment is rather short. The treatment failure rate was not associated with the patient volume among different centers [14]. In the present study, when comparing the tumor persistence rate between the initial 20 cases and a subsequent 20 cases in the focal ablation group, it was shown that there was no significant difference (76.5% vs. 70%, *p* = 0.659).

In the present study, the ablation area was expanded to one prostatic lobe; although the oncological results were encouraging, the impact on quality of life should not be ignored. Sexual function declined in the hemi-ablation group but increased in the focal ablation group, even though there were no statistically significant differences between the two groups. In a randomized study, 51 patients received focal ablation, and 55 patients received extended ablation for localized low–intermediate risk prostate cancer. The median follow-up time was 30 months. There were no significant differences in IPSS between the two groups across all time points. However, the focal ablation group showed significantly better sexual function (measured by IIEF-15 or Expanded Prostate Cancer Index Composite—sexual function domain) than the extended ablation group at 3-month and 6-month follow-ups [13]. It is worth mentioning that, during the study period, we performed few cases of whole-gland IRE ablation, but we have not reported on them. They had better oncological outcomes at the price of a significant increase in morbidity. The oncological control and the impact on quality of life should be balanced, and we therefore abandoned whole-gland ablation.

In the present study the hemi-ablation group bore a much longer time with a suprapubic catheter. A systematic review reported a catheterization time of 0–7 days after IRE treatment, and only one study reported 15.2 days of indwelling transurethral catheterization [12]. Although prolonged catheterization might reduce the risk of urinary retention, it may affect the patients’ quality of life.

An ablative treatment for localized prostate cancer is only as good as the detection of cancerous lesions and the technology used. Several approaches may address the limitations we are presently facing with ablative treatments. These may include enhanced imaging and innovative treatment protocols such as PSMA PET-CT and artificial intelligence (AI)-guided diagnosis and treatment.

In some studies, PSMA PET-CT has been shown to be superior to conventional imaging in the detection and localization of prostate cancer [21,22]. Two studies reported the application of PSMA PET-CT to correlate prostatic lesions with MRI and biopsy before IRE ablation [23,24], showing that PSMA PET-CT combined with MRI might have better value as a diagnostic approach for tumor localization before IRE.

Artificial intelligence (AI) applications present an unprecedented platform for improvement in the medical field including in the diagnosis, treatment, and prognostication of prostate cancer. In a recently published study, an AI system showed superiority to radiologists using Prostate Imaging-Reporting and Data System version 2.1 at detecting clinically significant prostate cancer and was comparable to the standard of care [25]. AI-based software could provide additional value in the ultrasound–MRI fusion targeted prostate biopsy and improve the diagnostic accuracy [26]. In the future, novel imaging and AI techniques may aid in the safe application and promotion of IRE-focal therapy.

The role of imaging and markers in the setting of post-IRE-focal therapy is poorly established, and there is limited evidence supporting clinical practice. MpMRI has shown excellent diagnostic value in treatment-naive patients; however, in the setting of post-IRE therapy, one study reported that mpMRI had a high negative predictive value (NPV) for detecting infield (91.4%) and whole-gland (86.7%) residual tumor, but the sensitivity was as low as 36% and 43.6%, respectively [27]. Apparently, the interpretation guidance for mpMRI assessment after focal therapy is lacking. Moreover, there is no consensus on the definition of treatment failure and the trigger for retreatment. In the future, more evidence is needed to further evaluate patient selection and modality options for retreatment considerations.

Our study has some limitations. First, the study size was relatively small; only 106 patients were included in two groups. Although we could reach statistically powered conclusions, it might not be adequately powered to detect small differences between the two groups. Second, this study was inherently retrospective in design. However, we obtained results in line with multi-center randomized studies [13,14]. Hence all patients followed a strict follow-up protocol providing a complete dataset for analysis. Furthermore, this study evaluated only the short-term oncological results. Given the objective of the present analysis, we decided to provide information on primary treatment outcomes, but we did not expand on the outcome of the retreatments. Finally, MRI-targeted fusion biopsy plus systematic biopsy was used in this study. Biopsy strategies vary among different IRE studies, and each approach may still have missed (clinically significant) prostate cancers. The strength of the present study is that this study unequivocally confirms that a larger treatment zone provides superior oncological outcomes without compromising mid–long term functional outcomes. Our earlier randomized controlled study compared a focal versus an extended ablation protocol [13,14], the latter with a treatment zone in between a focal ablation and a hemi-ablation. The present study signifies that better oncological outcomes will be obtained with a hemi-ablation over a focal or extended ablation in the presence of confirmed significant prostate cancer in one prostate lobe. Hence a treatment beyond hemi-ablation may further enhance oncological outcomes with a significant impairment of functional outcomes and with significant morbidity. Therefore we may safely conclude that the present study closes the search for the optimal treatment protocol for IRE ablation of localized prostate cancer.

## 5. Conclusions

Both IRE protocols achieved good urinary and sexual function outcomes and favorable short-term oncological control in men with localized low–intermediate risk prostate cancer. Hemi-ablation treatment showed significantly lower persistence of (clinically significant) prostate cancer compared with focal ablation.

## Figures and Tables

**Figure 1 cancers-17-02084-f001:**
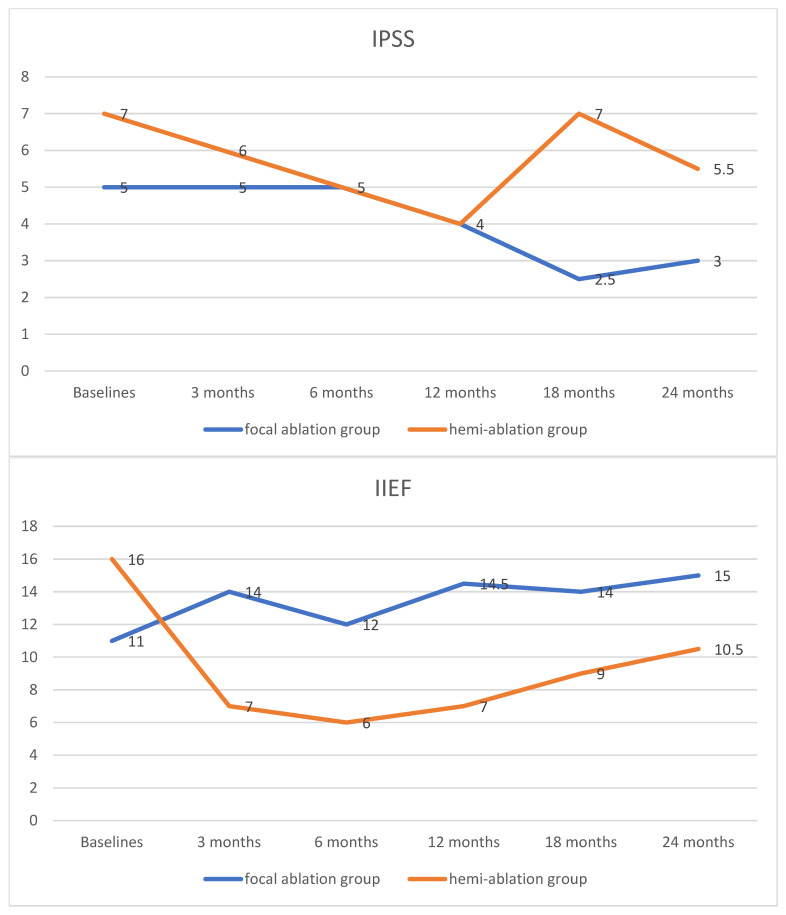
The change in quality of life measured with IPSS and IIEF during follow-up. IPSS: International Prostate Symptom Score; IIEF: International Index of Erectile Function.

**Table 1 cancers-17-02084-t001:** Baseline characteristics and perioperative outcomes.

	Focal Ablation Group (n = 40) (Median, IQR)	Hemi-Ablation Group (n = 66) (Median, IQR)	*p*-Value
Age (years)	68.5 (63.3–74.0)	71.0 (66.0–75.0)	0.138
PSA (ng/mL)	8.3 (5.6–11.6)	6.8 (5.4–9.1)	0.109
Prostate volume (mL)	42.0 (30.0–60.0)	44.0 (30.3–58.8)	0.537
Biopsy cores	21 (18–23)	22 (18–23)	0.963
ISUP			0.934
1	20	30	
2	16	30	
3	3	6	
4	1		
Number of electrodes	4 (4–4.8)	5 (5–5)	<0.001
Duration of catheterization	1 (0–2.8)	11 (8–16.8)	<0.001

ISUP: International Society of Urological Pathology; PSA: prostate specific antigen; IQR: interquartile range.

**Table 2 cancers-17-02084-t002:** The repeat biopsy results.

	Focal Ablation Group (n = 36)	Hemi-Ablation Group (n = 58)	*p*-Value
Negative, No. (%)	10 (27.8%)	40 (69.0%)	<0.001
Persistent tumor, No. (%)	26 (72.2%)	18 (31.0%)	
Persistent clinically significant tumor, No. (%)	9 (25%)	5 (8.6%)	0.003
Infield	16 (44.4%)	5 (8.6%)	
ISUP 1	5	3	
ISUP 2	4	0	
ISUP 3	0	1	
ISUP 4	1	1	
ISUP 5	1	0	
N/A	5	0	
Outfield	4 (11.1%)	12 (20.7%)	
ISUP 1	2	8	
ISUP 2	0	1	
ISUP 3	1	1	
N/A	1	2	
In- and outfield	6 (16.7%)	1 (1.7%)	
ISUP 1	2	0	
ISUP 2	1	1	
ISUP 5	1	0	
N/A	2	0	
Retreatment	21 (58.3%)	9 (15.5%)	<0.001
Radical prostatectomy	7	1	
Radiotherapy	7	2	
IRE	1	6	
ADT	1	0	
Radical prostatectomy +Radiotherapy	3	0	
IRE +Radiotherapy	2	0	

ISUP: International Society of Urological Pathology. IRE: irreversible electroporation. ADT: androgen deprivation therapy.

## Data Availability

The data that support the findings of our study are available from the corresponding author upon request.

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
