# Peer review of "Oncological and Functional Outcomes of Hemi-Ablation Versus Focal Ablation for Localized Prostate Cancer Using Irreversible Electroporation"

_cancers, 2025, doi:10.3390/cancers17132084_

Round 1
Reviewer 1 Report
Comments and Suggestions for Authors
The manuscript compares the focal irreversible electroporation (IRE)-ablation to the hemi-ablation for prostate cancer in a cohort of 106 patients. Hemi-ablation showed better outcomes to treat patients with low-intermediate risk of prostate cancer compared to focal IRE-ablation.
The introduction is easy to understand. Patients and Methods needs to be revisited. The discussion is well-written and supported by results.
I have provided minor comments for improvements to Methods and Results sections.
Minor revisions
Patients and Methods
Please, in the method section add the International Society of Urological Pathology (ISUP) grade groups and Gleason score utilized in this study.
Results
Line 102. The authors should include the terms “grade groups” to clarify that they are describing tumors staging according to the International Society of Urological Pathology (ISUP).
Table 1. The table should report the term grade to clarify the data are referring to tumor’s stage.
Line 107. The authors report IPSS and IIEF in Table 1 legend. However, IPSS and IIEF values are not reported in Table 1.
Figure 1. please, correct the legend spelling.
I hope the comments can help the authors to improve the manuscript.
Author Response
Comments 1: Patients and Methods
Please, in the method section add the International Society of Urological Pathology (ISUP) grade groups and Gleason score utilized in this study.
Response 1: Thanks for your kind advice. We have revised
Comments 2: Results
Line 102. The authors should include the terms “grade groups” to clarify that they are describing tumors staging according to the International Society of Urological Pathology (ISUP).
Response 2: Thanks for your kind advice. We have revised
Comments 3: Table 1. The table should report the term grade to clarify the data are referring to tumor’s stage.
Response 3: Thanks for your kind advice. I agree with this comment, but we have already clarified that in the manuscript, so it might not necessary to add this in the table. And it might compromise the simplicity of the table. Thanks
Comments 4: Line 107. The authors report IPSS and IIEF in Table 1 legend. However, IPSS and IIEF values are not reported in Table 1.
Response 4: Thanks for your kind advice. We have revised
Comments 5: Figure 1. please, correct the legend spelling
Response 5: Thanks for your kind advice. We have revised
Reviewer 2 Report
Comments and Suggestions for Authors
The article by Suberville, Zhang et al. described a set of assessments, including International Prostate Symptom Score (IPSS) for Oncological outcome and International Index of Erectile Function (IIEF) for Functional Outcomes, to compare Hemi Ablation (extended ablation) and Focal Ablation for Localized Prostate Cancer using Irreversible Electroporation (IRE).
This work was built upon prior publications, including:
- de la Rosette, J., Dominguez-Escrig, J., Zhang, K., Teoh, J., Barret, E., Ramon-Borja, J. C., ... & Laguna, P. (2023). A multicenter, randomized, single-blind, 2-arm intervention study evaluating the adverse events and quality of life after irreversible electroporation for the ablation of localized low-intermediate risk prostate cancer. The Journal of urology, 209(2), 347-353.
- Zhang, K., Teoh, J., Laguna, P., Dominguez-Escrig, J., Barret, E., Ramon-Borja, J. C., ... & de la Rosette, J. (2023). Effect of focal vs extended irreversible electroporation for the ablation of localized low-or intermediate-risk prostate cancer on early oncological control: a randomized clinical trial. JAMA surgery, 158(4), 343-349.
and offers additional information, especially the IPSS and IIEF over time between extended ablation and focal ablation as presented in Figure 1.
Even though similar comparisons have been made, including the above ones by the authors and a few other groups:
- Scheltema, M. J., van den Bos, W., de Bruin, D. M., Wijkstra, H., Laguna, M. P., de Reijke, T. M., & de la Rosette, J. J. (2016). Focal vs extended ablation in localized prostate cancer with irreversible electroporation; a multi-center randomized controlled trial. BMC cancer, 16, 1-9.
- [not the same IRE though] Wang, H., Xue, W., Yan, W., Yin, L., Dong, B., He, B., ... & Xu, C. (2022). Extended focal ablation of localized prostate cancer with high-frequency irreversible electroporation: a nonrandomized controlled trial. JAMA surgery, 157(8), 693-700.
The reviewer finds this article informative, additional description and details (if applicable) can improve the clinical significance of this article, including:
- Inclusion and exclusion criteria in “2.1. Study Design” and “2.2. Patient Selection”
- Perioperative characteristics, for instance the Biopsy Gleason Score (3+3, 3+4, 4+3 etc.), Clinical stage (T1c, T2a, T2b, T2c etc.).
- Expanded Prostate Cancer Index (EPIC) or Visual Analogue Scale (VAS)
- The data range (Scores, standard deviation) in Figure 1.
- IPSS and IIEF for subgroups, for example based on ISUP 1 or Infield vs outfield.
Author Response
Comments 1: Inclusion and exclusion criteria in “2.1. Study Design” and “2.2. Patient Selection”
Perioperative characteristics, for instance the Biopsy Gleason Score (3+3, 3+4, 4+3 etc.), Clinical stage (T1c, T2a, T2b, T2c etc.).
Expanded Prostate Cancer Index (EPIC) or Visual Analogue Scale (VAS)
The data range (Scores, standard deviation) in Figure 1.
IPSS and IIEF for subgroups, for example based on ISUP 1 or Infield vs outfield.
Response 1: Thanks for your kind advice.
The biopsy Gleason score was shown in table 1,2, the data range of IPSS and IIEF was shown in the supplemental table 3
Because of the retrospective nature of the study, some data are not available, such as clinical stage, EPIC, VAS. In our study, all the patients with localized prostate cancer and who were willing to receive IRE focal treatment were included, there was no specific inclusion and exclusion criteria. We also believe all the missing data would improve our manuscript. Thanks again